# Application of alternative nonlinear models to predict growth curve in partridges

**Navid Ghavi Hossein-Zadeh** ⬤ *

Department of Animal Science, Faculty of Agricultural Sciences, University of Guilan, Rasht, Iran

* nhosseinzadeh@guilan.ac.ir, navid.hosseinzadeh@gmail.com

## Abstract

This study aimed to describe the growth pattern in partridges using nonlinear models. Eight nonlinear mathematical functions (Bridges, Janoschek, Richards, Schumacher, Morgan, Lomolino, Sinusoidal, and Weibull) were used. The parameters of nonlinear models were estimated by fitting the models to partridge body weight records using the NLIN and MODEL procedures in the SAS program. Model performance was assessed and model behavior was examined during the process of fitting nonlinear regression curves. The overall goodness of fit of each model to various data profiles was assessed using the adjusted coefficient of determination, root mean square error, Akaike's information criterion, and Bayesian information criterion. The adjusted coefficient of determination values for each model are generally high, indicating that the models fit the data well overall. Based on goodness of fit criteria, the Morgan model was found to be the most appropriate function for fitting the growth curve of male and female partridges. Furthermore, the Lomolino model had the worst fit to the growth curves of male and female partridges. While the predictions of the final body weight from all the models were good, the Morgan function outperformed the others in this regard. Based on the first derivative of the Morgan model, the absolute growth rates for male and female partridges as a function of time revealed that these values gradually increased with increasing age until 42 and 35 days of age, respectively, and then declined. The Morgan function is a useful replacement for conventional growth functions when describing the growth curves of different partridge breeds.

## Introduction

In an attempt to duplicate the advantages of animals raised in the wild, like increased productivity and better flavor, humans have had difficulty raising hunting animals in controlled environments. Scientists are searching for novel approaches to raise various game animals for meat as a result of this struggle. The two main game animals that are raised intensively these days are partridges and pheasants [1–2]. Partridges are game birds found in the wild and are primarily raised for meat production, hunting tourism, and environmental conservation [2–4]. When they reach a certain weight and age, partridges bred for hunting tourism or maintaining natural balance are released into hunting areas or the wild [5]. Partridges play a key role in the rural economy, particularly in regions where hunting tourism contributes significantly to local income. In these areas, farms specifically breed

**Data availability statement:** All relevant data are within the manuscript and its Supporting Information files.

**Funding:** The author(s) received no specific funding for this work.

partridges for release into hunting reserves, contributing to both conservation efforts and economic sustainability. Partridges are ideal for commercial production due to their rapid growth, high productivity, efficient feed conversion, and exceptional meat quality. Partridges are considered a premium source of protein in the market due to their lean, flavorful, and nutritious meat, which has led to a global increase in demand for wild birds [6]. In captivity, partridges are an adaptable breed that work well in commercial production [7]. This has led to an increase in the number of businesses that raise partridges for meat. Partridges have similar breeding and feeding practices to other species of poultry [8]. These characteristics have led to an increase in partridge breeding and current research aims to improve productivity and growth characteristics to meet the growing demand for wild birds in the food and tourism sectors.

The potential of partridges for further domestication and genetic development also allows their productivity to be maximized and strengthens their importance in industrial poultry farming systems.

Growth is defined as changes in live weight and proportionate body component growth influenced by environmental factors and genotype; growth curves are defined as changes in growth that occur over time [9]. Growth curves are the mathematical representation of growth, and their main use is to summarize data collected at various ages. Despite their difficulty, there are a few parameters that can be understood in the context of biology [10–11]. Growth curves can be used to find the best age to slaughter, identify a measurable growth trait, learn more about the health of the living thing, and assess how selection has affected the growth curve's parameters [12–13]. Comprehending the partridge's growth curve parameters could aid scientists in initiating animal breeding initiatives [5].

When examining the growth curve of farm animals, there are a number of reasons to think about utilizing alternative models rather than traditional ones. The complex and non-linear growth trajectories seen in farm animals may not be sufficiently captured by conventional models, which frequently assume linear growth patterns. Different models are more capable of explaining differences in growth rates at various developmental stages. Alternative models allow for more customization and flexibility when fitting growth data, making it possible to capture distinct growth patterns unique to various farm animal species or breeds. By accounting for variables that can affect growth rates, such as genetic influences, environmental factors, and feed quality, alternative models may offer a more accurate depiction of growth trajectories. Alternative models might produce more accurate growth performance forecasts, empowering farmers to choose the best breeding selection, feed plans, and general management techniques [14]. In general, alternative models can provide a more thorough and nuanced approach to analyzing farm animal growth curves than conventional models, which can enhance insights and decision-making in animal production systems [14]. Several asymptotic-mechanistic models have been developed using various mathematical functions to simulate livestock growth. These models incorporate functions based on the biological growth of the animal [15]. Parameterizing mathematical models of partridge growth and assessing alternative management and feeding strategies are necessary for optimizing partridge production systems. Nevertheless, only a few studies have described the growth curves of different partridge breeds [5,7,8,16–20], and most of these studies used traditional nonlinear models such as Brody, Gompertz, logistic, Von Bertalanffy, and Richards. Although these models are well established and used successfully to describe the growth of various livestock species, they may not fully capture the complexity and variability of growth patterns, particularly in species such as partridges. The key limitation of these traditional models is their relatively rigid structure, which may not take into account the different growth phases or environmental interactions that are important in more nuanced growth studies. Consequently, there are very few studies

on partridge growth curves in the literature, particularly when using alternative models such as Schumacher, Morgan, Lomolino, Weibull, and sinusoidal. Therefore, this study aimed to evaluate the suitability of eight nonlinear models (Bridges, Janoschek, Richards, Schumacher, Morgan, Lomolino, Sinusoidal, and Weibull) to describe the growth pattern of partridges to incorporate alternative functions into the study and production of partridges. By evaluating a greater variety of nonlinear models, this study fills the gaps left by traditional models. This method provides greater flexibility in documenting the different growth stages of partridges and allows for a more thorough study of growth patterns. By comparing alternative models, this study provided a clearer understanding of which models are best for predicting partridge growth, which could improve breeding and management strategies. Furthermore, this study contributed to the literature by expanding the set of tools available for modeling partridge growth, going beyond the limitations of previous work that focused exclusively on traditional models.

## Materials and methods

### Ethics statement

This study utilizes data previously collected and published in existing literature. Given that the data is publicly available, no additional ethics approval was required for this study.

### Data sources

Records of body weight of chukar partridges (*Alectoris chukar*) from hatching to 140 days of age are from the study of Iqbal et al. [8] and used in the current study (S1 Table). A total of 72 male and 108 female partridge chicks were weighed individually and had their wings banded. Every week, a digital scale with a sensitivity of 0.01 g was used to weigh them. Detailed information on the housing and management of the birds can be found in a study reported by Sariyel et al. [5]. Sariyel et al. [5] compared the goodness of fit of the Brody, Gompertz, Logistic, and von Bertalanffy growth curve models in this data set. Also, Iqbal et al. [8] fitted the Gompertz, Brody, Logistic, and von Bertalanffy growth models on the same data set using a Bayesian approach.

### Nonlinear models

Table 1 displays the nonlinear models that were utilized to explain the growth curves. To model the relationship between body weight and age, the following models were fitted to the data: Bridges, Janoschek, Schumacher, Richards, Morgan, Lomolino, sinusoidal, and Weibull.

### Statistical analysis

The parameters of eight nonlinear models were estimated by independently fitting the models to partridge body weight records using the NLIN and MODEL procedures in SAS 9.1 [29]. Iteration techniques for fitting non-linear functions were based on the Gauss-Newton method. Before beginning this process, the NLIN and MODEL procedures evaluates the starting value specifications for the parameters. The initial values of the parameters had to be supplied for the iterative process to work. The final estimates were unaffected by the initial values that were chosen.

Using the adjusted coefficient of determination ( $R_{adj}^2$ ), residual standard deviation or root means square error (RMSE), Durbin-Watson statistic (DW), Akaike's information criterion (AIC), and Bayesian information criterion (BIC), the quality of the predictions, or goodness of fit, was assessed for each model.

**Table 1. Functional forms of nonlinear models for describing the growth curve of partridge.**

| Model | Equation | Number of parameters | Reference |
|---|---|---|---|
| Bridges | $y = W_0 + a\left(1 - e^{-kt^m}\right) + \varepsilon$ | 4 | [21] |
| Janoschek | $y = a - \left((a - W_0)e^{-kt^m}\right) + \varepsilon$ | 4 | [22] |
| Richards | $y = \dfrac{a}{\left(1 - be^{-kt}\right)^{\frac{1}{m}}} + \varepsilon$ | 4 | [23] |
| Schumacher | $y = \dfrac{ab^2 k}{(t+b)^2}e^{\left(\frac{bkt}{t+b}\right)} + \varepsilon$ | 3 | [24] |
| Morgan | $y = ab^k k\dfrac{t^{k-1}}{\left(t^c + b^k\right)^2} + \varepsilon$ | 3 | [25] |
| Lomolino | $y = \dfrac{a}{1 + b^{\log\left(\frac{k}{t}\right)}} + \varepsilon$ | 3 | [26] |
| Weibull | $y = a - (a-b)e^{-\left(\frac{k-1}{k}\right)\left(\frac{t}{IP}\right)^k} + \varepsilon$ | 4 | [27] |
| Sinusuidal | $y = y_0 + a \times \sin\left(\dfrac{2\pi t}{b} + k\right) + \varepsilon$ | 4 | [28] |

y= represents body weight at age t (day); $\varepsilon$ = Random error; a= represents asymptotic weight, which is interpreted as mature weight; and b= is an integration constant related to initial animal weight. The value of b is defined by the initial values for y and t; k= is the maturation rate, which is interpreted as weight change in relation to mature weight to indicate how fast the animal approaches adult weight; m= is the parameter that gives shape to the curve by indicating where the inflection point occurs; IP= is inflection point; $W_0$= is initial body weight. For sinusoidal function, *a* is the amplitude, $y_o$ is the vertical offset and *k* is the phase shift. This sinusoidal function is periodic with period *b*. Also, for sinusoidal function, $W_0 = y_0 + a \times \sin(k)$, and final weight is calculated by $a + y_0$.

The following formula was used to calculate $R^2_{adj}$ :

$$R^2_{adj} = 1 - \left[\frac{(n-1)}{(n-p)}\right]\left(1 - R^2\right)$$

Where, the multiple coefficient of determination is denoted by $R^2$ ( $R^2 = 1 - \dfrac{RSS}{TSS}$ ), n denotes the number of observations (data points), p denotes the number of parameters, RSS stands for residual sum of squares, and TSS stands for total sum of squares. The $R^2$ value is used to determine how much of the total variation around the mean of the trait can be attributed to the growth curve model. The $R^2$ always falls within the range of 0–1, with the model considered satisfactory when $R^2$ is close to 1.

RMSE is a form of standard deviation that is calculated in the following way:

$$RMSE = \frac{RSS}{\sqrt{n - p - 1}}$$

The residual sum of squares (RSS) represents the error in the data, with n being the number of observations and p being the number of parameters in the equation. The RMSE value is a critical factor in evaluating the appropriateness of growth curve models, with the most favorable model being the one with the lowest RMSE.

To assess the normality of the residuals of the fitted nonlinear models, the Shapiro-Wilk test was used. This test is commonly used to examine whether the residuals follow a normal distribution. A significant *P*-value (*P*<0.05) indicates deviations from normality. To check the assumption of constant variance in the residuals, the White's test was also used. White's test is a common statistical test for heteroscedasticity that evaluates whether the variance of the residuals remains stable across different levels of the independent variable. A non-significant result of the White's test ($P > 0.05$) would indicate that the homoscedasticity assumption is met, supporting the validity of the regression models. The Durbin-Watson (DW) statistic was utilized to investigate the residuals from the regression analysis for the existence of autocorrelation. Given the presence of autocorrelated residuals, it is possible that the function is inappropriate for the given data. The DW statistic has values between 0 and 4. Autocorrelation is absent when the value is close to two, positive when the value is close to zero, and negative when the value is close to four [30]. The significance of the DW values was tested using R software version 4.4.2. A significant autocorrelation in the residuals ($P < 0.05$) would indicate that the model may not adequately capture the temporal structure of the data. The following formula was used to determine DW:

$$DW = \frac{\sum\limits_{t}^{n} \left(e_t - e_{t-1}\right)^2}{\sum\limits_{t=1}^{n} e_t^2}$$

Where, $e_t$ denotes the residual at time e, and $e_{t-1}$ presents residual at time t-1.

Using the following equation [31], AIC was calculated:

$$AIC = n \times \ln\left(RSS\right) + 2p$$

The AIC statistic is useful when comparing models with different levels of complexity because it adjusts the RSS based on the model's parameter count. A better fit is indicated by a lower AIC number value in the model comparison.

BIC integrates maximum likelihood (data fitting) and model selection by penalizing the (log) maximum likelihood with a term related to model complexity:

$$BIC = n\ln\left(\frac{RSS}{n}\right) + p\ln\left(n\right)$$

When comparing models, a lower BIC number denotes a better fit. In general, AIC and BIC are good criteria for comparing models that have different numbers of parameters.

Once the optimal function was chosen, the absolute growth rate, or AGR, was calculated using the function's first derivative with respect to time ($\frac{\partial y}{\partial t}$). AGR or average growth rate of animals per unit time is a useful tool for calculating the average growth rate of a population. In this case, this indicates the approximate daily weight gain during a growth phase [32]. In addition, the inflection point (IP) was calculated using the second derivative of the best function with respect to time [33–34]. As stated by Mischan et al. [33], IP is the value at which the growth model's second derivative is set to zero. The first and second derivatives of the best function were calculated using R software version 4.4.0.

## Results

### Model parameters and goodness of fit statistics

Table 2 displays estimated parameters of the nonlinear growth models for male and female partridges. In Table 3, goodness of fit statistics for the eight growth models fitted to body

**Table 2. Parameter estimates for the different growth models in partridge[*].**

| Item | Parameter[**] | Model | | | | | | | |
|------|-----------|-------|-----------|----------|------------|--------|----------|--------|------------|
| | | Bridges | Janoschek | Richards | Schumacher | Morgan | Lomolino | Weibull | Sinusuidal |
| Male | $y_0$ | – | – | – | – | – | – | – | 212.69 |
| | $W_0$ | 12.69 | 12.69 | – | – | – | – | – | – |
| | a | 539.49 | 552.19 | 576.81 | 86.03 | 171817.4 | 725.79 | 552.22 | 289.62 |
| | b | – | – | 0.6545 | 53.8997 | 232.5617 | 42.6009 | 12.6847 | 365.27 |
| | k | 0.001021 | 0.001021 | 0.021987 | 0.158805 | 2.260343 | 79.41522 | 1.58708 | 5.4617 |
| | IP | – | – | – | – | – | – | 40.94699 | – |
| | m | 1.587243 | 1.587238 | -0.26496 | – | – | – | – | – |
| | $y_0$ | – | – | – | – | – | – | – | 154.33 |
| | $W_0$ | 10.29 | 10.29 | – | – | – | – | – | – |
| Female | a | 452.90 | 463.19 | 481.65 | 81.70 | 146162.2 | 586.25 | 463.20 | 273.82 |
| | b | – | – | 0.7414 | 51.8563 | 230.3876 | 42.4601 | 10.2862 | 394.6870 |
| | k | 0.001439 | 0.001439 | 0.022655 | 0.161405 | 2.20242 | 70.29801 | 1.53370 | 5.6778 |
| | IP | – | – | – | – | – | – | 35.80403 | – |
| | m | 1.533751 | 1.533749 | -0.33003 | – | – | – | – | – |

[*]Body weight records were obtained from Iqbal et al. [8].

[**]Model parameters are defined in Table 1.

**Table 3. Comparing goodness of fit for different growth curves in partridge[*].**

| Item | Statistics | Model | | | | | | | |
|------|-----------|-------|-----------|----------|------------|--------|----------|--------|------------|
| | | Bridges | Janoschek | Richards | Schumacher | Morgan | Lomolino | Weibull | Sinusuidal |
| Male | $R^2_{adj}$ | 0.9973 | 0.9973 | 0.9971 | 0.9973 | 0.9975 | 0.9962 | 0.9973 | 0.9973 |
| | DW | 1.28 | 1.28 | 1.21 | 1.23 | 1.36 | 0.93 | 1.28 | 1.39 |
| | P-value for DW test | 0.023 | 0.023 | 0.014 | 0.017 | 0.043 | 0.002 | 0.023 | 0.042 |
| | P-value for Shapiro-Wilk test | 0.270 | 0.270 | 0.418 | 0.036 | 0.476 | 0.228 | 0.271 | 0.077 |
| | P-value for White's test | 0.196 | 0.197 | 0.302 | 0.110 | 0.191 | 0.541 | 0.196 | 0.058 |
| | RMSE | 8.96 | 8.96 | 9.34 | 8.92 | 8.55 | 10.70 | 8.96 | 8.89 |
| | AIC | 159.58 | 159.58 | 161.33 | 158.60 | 156.84 | 166.27 | 159.58 | 159.26 |
| | BIC | 99.83 | 99.83 | 101.57 | 97.80 | 96.04 | 105.46 | 99.83 | 99.50 |
| Female | $R^2_{adj}$ | 0.9973 | 0.9973 | 0.9972 | 0.9972 | 0.9973 | 0.9964 | 0.9973 | 0.9967 |
| | DW | 1.31 | 1.31 | 1.25 | 1.18 | 1.32 | 0.97 | 1.31 | 1.25 |
| | P-value for DW test | 0.026 | 0.026 | 0.018 | 0.012 | 0.030 | 0.004 | 0.026 | 0.018 |
| | P-value for Shapiro-Wilk test | 0.261 | 0.261 | 0.529 | 0.013 | 0.244 | 0.462 | 0.262 | 0.344 |
| | P-value for White's test | 0.136 | 0.136 | 0.220 | 0.085 | 0.099 | 0.374 | 0.136 | 0.041 |
| | RMSE | 7.55 | 7.55 | 7.78 | 7.75 | 7.64 | 8.79 | 7.55 | 8.42 |
| | AIC | 152.41 | 152.41 | 153.67 | 152.71 | 152.09 | 157.99 | 152.41 | 156.99 |
| | BIC | 92.65 | 92.65 | 93.91 | 91.91 | 91.28 | 97.19 | 92.65 | 97.24 |

[*]Body weight records were obtained from Iqbal et al. [8]

: Adjusted coefficient of determination; RMSE: Root means square error; DW: Durbin–Watson; AIC: Akaike information criteria; BIC: Bayesian Information Criteria

weight records are also presented. All models provided high $R^2_{adj}$ values, and although there were minimal differences in $R^2_{adj}$ values between the models, the Morgan model had the highest $R^2_{adj}$ value for male partridges, and the Bridges, Janoschek, Morgan, and Weibull

models had the highest values for female partridges. However, the Lomolino model produced the smallest values of $R^2_{adj}$ for male and female partridges. DW values ranged from 0.93 (for Lomolino) to 1.39 (for sinusoidal) in male partridges and from 0.97 (for Lomolino) to 1.32 (for Morgan) in female partridges. Therefore, positive autocorrelation was observed for all models ($P < 0.05$). For male and female partridges, with the exception of the Schumacher model, other functions had a normal residual distribution (P>0.05). The results of White's test examining the assumption of constant variance in the residuals confirmed residual homogeneity for all models in male partridges (P> 0.05), but the results showed that, with the exception of the sine model, other models provided residual homogeneity in female partridges (Table 3). For male partridges, the Morgan and Lomolino models provided the lowest and highest RMSE values, respectively. However, the Bridges, Janoschek, and Weibull models provided the lowest RMSE values for female partridges, but the Lomolino function had the highest value. Moreover, Morgan function had the lowest AIC and BIC values for male and female partridges, but the Lomolino model had the highest AIC and BIC values. Therefore, the Morgan model was determined to be the best function for fitting the growth curve of male and female partridges. After Morgan model, the Schumacher and sinusoidal functions provided the best fit of growth curve in male pertridges, respectively. But, after Morgan model, Schumacher model provided the best fit of growth curve in female partridges. Moreover, the Lomolino model had the worst fit to the growth curve of male and female partridges.

## Comparison of predicted growth curves

According to Iqbal et al. [8], partridge body weights were found to increase with age during the study period, with males becoming noticeably heavier than females after 140 days of age. For male and female partridges, the Lomolino, Morgan, and sinusoidal functions provided underestimated initial body weights. However, other models predicted initial body weights close to the actual value. There existed magnitude differences between the various functions for the final or asymptotic body weights. All models provided a good prediction of final body weight, but the Morgan function provided the best prediction of final body weight compared to other models. In general, initial and final body weights predicted by different models were typically greater for male partridges compared with female partridges. Fig 1 shows predicted body weights for male and female partridges as a function of age, calculated using different nonlinear models.

## First and second derivatives of the growth curve

Fig 2 displays the AGR values for male and female partridges as a function of time, based on the first derivative of the Morgan model. For male and female partridges, AGR values progressively rose with increasing age until 42 and 35 days of age, and then declined, respectively. For male and female partridges, based on the second derivative of the Morgan function, the IP was determined to be 42 and 35 days old, respectively.

## Discussion

This study presents a novel approach to modeling the growth curve of partridges by applying a wider range of nonlinear models than previously investigated, including less common models such as the Morgan, Lomolino, and sinusoidal functions. While traditional models such as Gompertz and logistic are widely used in poultry species, this study evaluates eight models using a comprehensive set of fitting criteria to provide a robust comparison. Traditional growth models are typically those that are widely used in livestock growth research and have proven themselves over many years. A relatively fixed biological interpretation of growth

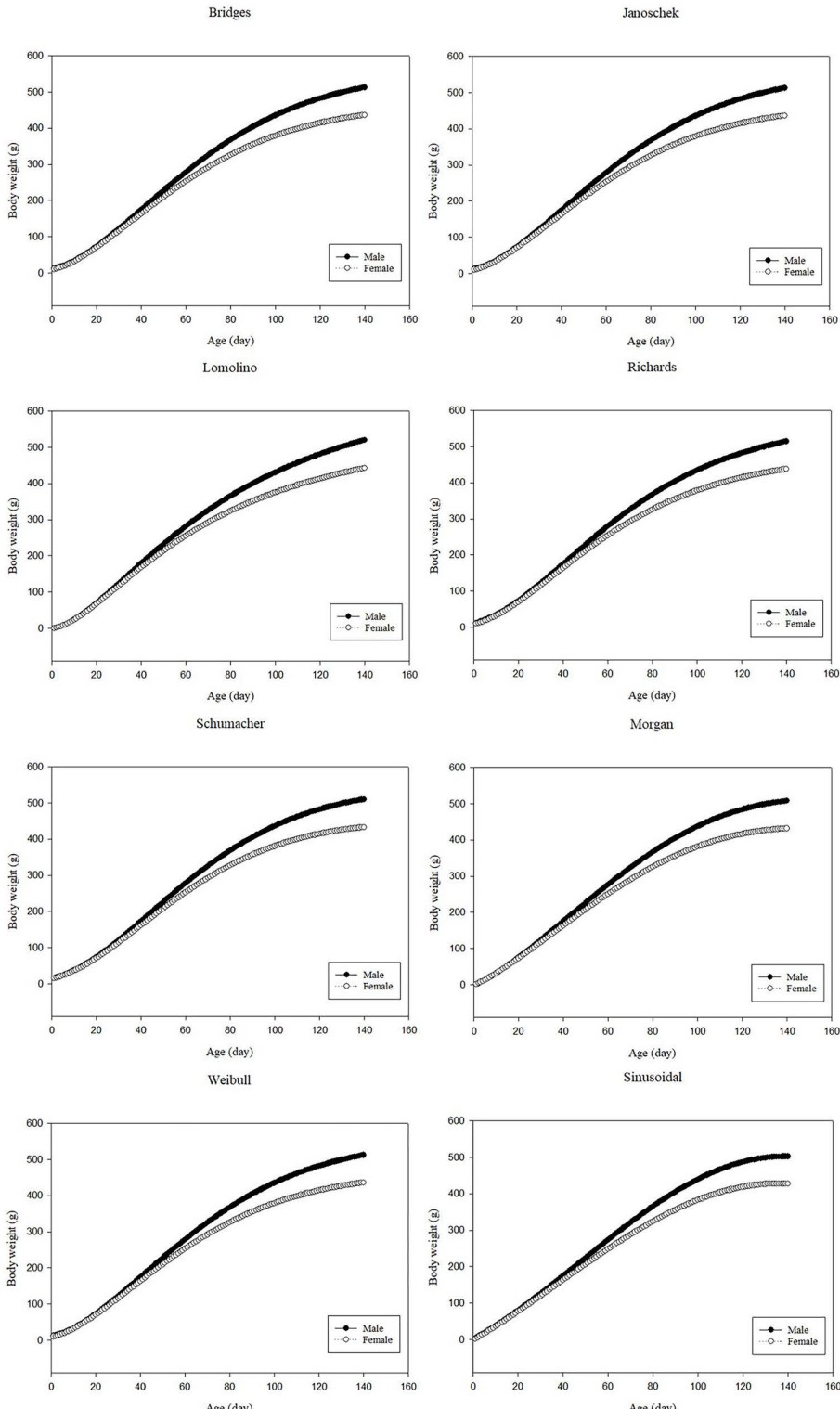

**Fig 1. Predicted body weights as a function of age, determined using different nonlinear models for male and female partridges.**

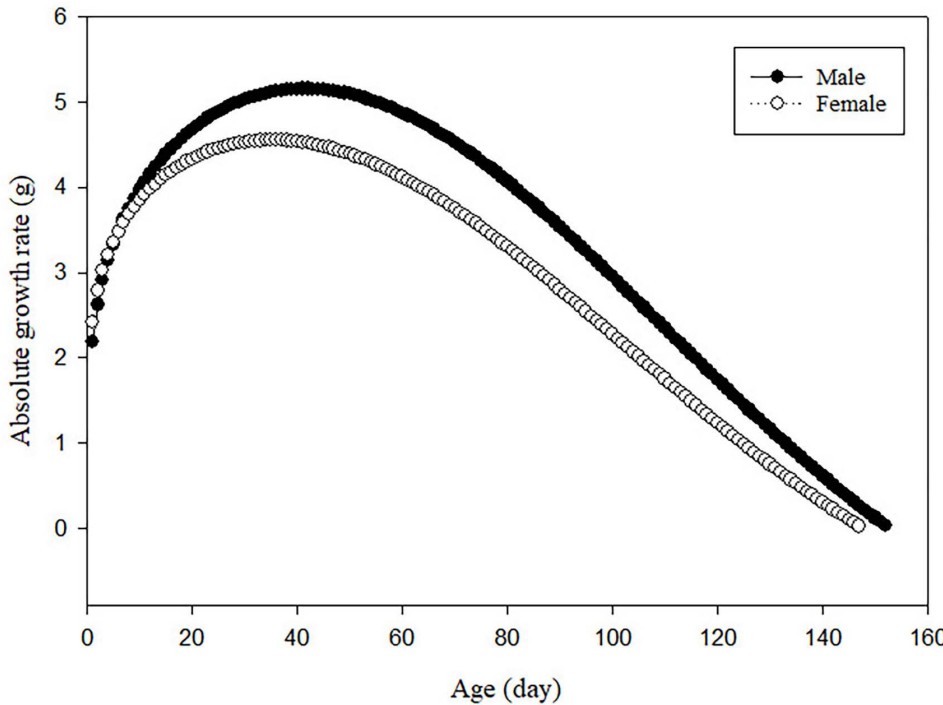

**Fig 2. Absolute growth rate (AGR) as a function of age for male and female partridges based on the Morgan model.**

phases, fewer parameters and simpler mathematical structures are some of the well-known features of these models (such as the Brody, Gompertz, Logistic, and von Bertalanffy models). Because of their widespread use and ease of use, they are commonly referred to as "conventional". In contrast, modern growth models (sometimes referred to as alternative or advanced models) are those that have been developed or adapted more recently to handle more complex or species-specific growth patterns. These models (such as the Morgan, Lomolino, Schumacher, and Weibull) tend to offer greater flexibility by incorporating additional parameters or more sophisticated mathematical functions. Modern models can better take nonlinear and irregular growth pattern into account, enabling more individual adaptation to species with complex or variable growth stages. The results showed that the Morgan model provided the most accurate fit for both male and female partridges, an important contribution that advances growth modeling for this species. By identifying this model as the best alternative to traditional models, the study fills a gap in the existing literature and provides a more precise tool for predicting growth dynamics. Additionally, this research extends the application of growth models beyond commercial production to conservation efforts and highlights the dual potential for optimizing breeding programs and managing game bird populations. Thus, the study makes a significant contribution by improving the accuracy of growth predictions in partridges and expanding the scope of growth modeling in both agricultural and ecological contexts. By maximizing breeding and release tactics and ensuring that birds are released into the wild at the best time for their survival, growth models contribute to partridge conservation. To improve bird health, they provide guidance on optimal rearing techniques, such as nutritional management. Growth models ensure a balanced gene pool through targeted breeding, which further promotes genetic diversity. They also enable adaptive management in response to changing environmental conditions and balance conservation goals with business

objectives, for example in agricultural environments, ensuring both sustainability and species productivity. The use of body weight data from a previous study (Iqbal et al. [8]) is one of the potential limitations of the study, potentially limiting the applicability of the results to different partridge populations or environmental conditions. Even though the sample size was adequate, it could still limit wider use. Although the Morgan model best fits the data, it has not been tested for other species or conditions. Furthermore, the study's dependence on specific management and husbandry conditions (according to Sariyel et al. [5]) could limit its suitability for use in different rearing situations. Although the accuracy of the body weight measurements is sufficient, there may be slight fluctuations. Finally, the eight nonlinear models tested may not cover all possible growth patterns, particularly under more complex or variable conditions.

Accurately fitting growth curves and analyzing their parameters are essential for improving breeding and production in livestock and poultry [16]. This study aimed to identify the best model to describe partridge live weight data, which is critical for understanding growth dynamics, optimizing feeding strategies, and determining the ideal age for slaughter. Additionally, the growth curve can help evaluate genetic quality, nutrition, and management practices. The Morgan model emerged as the most appropriate function, based on its superior fit and ability to capture the growth trajectory of both male and female partridges. While all models performed well, the Morgan model's parameters best represented the biological processes of growth, though the choice of model should always consider both statistical fit and biological relevance [35–36].

Thus far, nonlinear models have been used by some researchers to examine the growth curves of various partridge breeds. Cetin et al. [17] used the growth profiles of male and female partridges to compare the growth models (Gompertz, Logistic, and Richards). Their results showed that the Gompertz model more accurately covers the data for both male and female partridges. Also, Tholon et al. [18] used the Gompertz model in partridge research. Tholon and Queiroz [19] concluded that the Gompertz model was the most effective model to describe tinamous partridge growth due to the highest coefficients of determination, easy convergence, lower mean square predicted error, and simplicity of biological parameter interpretation. Balcioglu et al. [7] compared the growth data of male and female partridges using Gompertz, logistic, and von Bertalanffy models. Aourir et al. [20] investigated Barbary partridges (*Alectoris barbara*) using the Gompertz model. Sariyel et al. [5] analyzed growth data of partridges and concluded that the Gompertz model was most effective in determining the growth pattern. Wen et al. [16] found that the Weibull-type model explained partridge live weight data the best. Iqbal et al. [8] found that the von Bertalanffy model offered the best growth curve fit when utilizing a Bayesian approach to describe the growth of Chukar partridges. Compared with the results of Sariyel et al. [5] and Iqbal et al. [8], the results of the current study suggest a better fit of alternative nonlinear models used to describe the growth curve of partridges based on goodness of fit criteria. The Morgan model has a flexible inflection point and removes the limitation of fixed inflection point growth models such as the Gompertz equation.

The varying degrees of goodness of fit among different breeds of partridge can be attributed to differences in growth curve properties. The discrepancies in model fit seen in various studies may be the result of variations in factors such as animal breeds, body weight records, data points, and mathematical forms of the models. These variations in growth curves can be influenced by both genetic and environmental factors.

The detection of positive autocorrelations, also known as serial correlation, among the residuals in all models suggests underlying patterns not fully captured by the nonlinear growth models used in this study [30]. These autocorrelations could stem from factors such as

unmodeled genetic influences, environmental variables, or time-dependent trends that affect partridge growth. Positive autocorrelation indicates that growth dynamics may be influenced by latent variables, underscoring the complexity of accurately modeling biological growth processes in this species. Importantly, the presence of autocorrelation in the residuals signals potential model misspecification or the omission of key growth-related variables, a challenge that has been similarly observed in other poultry species. This study highlights the need for advanced modeling techniques, such as the inclusion of first-order autoregressive (AR (1)) models, to account for residual dependence. Implementing AR (1) corrections could improve the precision of parameter estimates and enhance the reliability of growth predictions by addressing serial dependencies in the data [37]. By directly confronting autocorrelation, the study offers a path forward for refining growth curve modeling in partridges and possibly other poultry species, emphasizing the importance of selecting models that balance fit with biological accuracy.

The analysis of AGR provides a dynamic view of how partridges grow over time, revealing periods of rapid growth as well as slower phases. Rather than relying solely on static weight measurements, AGR allows for a deeper understanding of growth trajectories, helping to identify critical growth stages such as the inflection point (IP), where growth peaks before tapering off [14,38]. This study found that both male and female partridges reached their maximum growth rates at the IP, after which the rate of weight gain steadily declined. This trend highlights the need for targeted management interventions at key growth stages, such as adjusting feed or monitoring health, to optimize weight gain during periods of rapid growth. The later occurrence of the IP in males compared to females underscores the impact of sexual dimorphism on growth patterns in partridges. Males typically grow larger and take longer to reach full maturity, which may explain the delayed peak in their growth rate. Recognizing these differences can guide more precise nutritional strategies, allowing farmers to tailor feeding schedules to the specific needs of males and females, ultimately improving feed efficiency and reducing waste [39]. By tracking changes in AGR, producers can better understand the growth potential of their flock and make informed decisions to maximize productivity, particularly during critical growth periods when management adjustments can have the greatest impact.

## Conclusions

In this study, the performance of eight nonlinear models in predicting the growth curves of partridges was successfully evaluated, with the aim of identifying the most suitable model for this species. Among the models analyzed, the Morgan model was found to be the most suitable for describing the growth patterns of both male and female partridges due to its superior goodness of fit criteria (e.g., RMSE, AIC, BIC, and $R_{adj}^2$). The Morgan model's accuracy in predicting growth curves provides valuable insights into optimizing partridge management and breeding, enabling more precise estimates of growth rates and final body weight. These results highlight the importance of selecting a robust and flexible model that can be adapted to different growth profiles in commercial and conservation environments. However, it is critical to validate the performance of the Morgan model with larger data sets that are measured more frequently to ensure its accuracy and reliability under a wider range of conditions. Additionally, future research should explore the application of this model to different breeds and environmental conditions to further validate its versatility and usefulness in poultry growth studies.

## Supporting information

**S1 Table.  Data sets for modeling the growth curve in male and female partridges.**
(DOCX)

## Author contributions

**Conceptualization:** Navid Ghavi Hossein-Zadeh.

**Formal analysis:** Navid Ghavi Hossein-Zadeh.

**Investigation:** Navid Ghavi Hossein-Zadeh.

**Methodology:** Navid Ghavi Hossein-Zadeh.

**Project administration:** Navid Ghavi Hossein-Zadeh.

**Supervision:** Navid Ghavi Hossein-Zadeh.

**Validation:** Navid Ghavi Hossein-Zadeh.

**Visualization:** Navid Ghavi Hossein-Zadeh.

**Writing – original draft:** Navid Ghavi Hossein-Zadeh.

**Writing – review & editing:** Navid Ghavi Hossein-Zadeh.

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
