## [Decision Letter · Decision Letter 0]

15 Sep 2024

PONE-D-24-33283Application of alternative nonlinear models to predict growth curve in partridgesPLOS ONE

Dear Dr. Ghavi Hossein-Zadeh,

Thank you for submitting your manuscript to PLOS ONE. After careful consideration, we feel that it has merit but does not fully meet PLOS ONE’s publication criteria as it currently stands. Therefore, we invite you to submit a revised version of the manuscript that addresses the points raised during the review process.

We look forward to receiving your revised manuscript.

Kind regards,

Wenpeng You

Academic Editor

PLOS ONE

Journal Requirements:

Reviewers' comments:

Reviewer's Responses to Questions

**Comments to the Author**

1. Is the manuscript technically sound, and do the data support the conclusions?

Reviewer #1: Yes

Reviewer #2: Yes

2. Has the statistical analysis been performed appropriately and rigorously? 

Reviewer #1: Yes

Reviewer #2: Yes

3. Have the authors made all data underlying the findings in their manuscript fully available?

Reviewer #1: Yes

Reviewer #2: Yes

4. Is the manuscript presented in an intelligible fashion and written in standard English?

Reviewer #1: Yes

Reviewer #2: Yes

5. Review Comments to the Author

Reviewer #1: The manuscript analyzes growth curves in poultry using fixed non-linear models. It is not new or innovative, but it uses functions that are not commonly seen or discussed in the literature, which makes it interesting. A suggestion would be to use non-linear in mixed models (proc nlimixed, in SAS).

In the introduction, it would be interesting to briefly highlight the commercial importance of the animal species being evaluated.

The metrics used for comparisons between models are pertinent and provide a good basis for discussing the paper.

I would have preferred the conclusion to be more direct in relation to the objective of the paper, I think it loses a lot in repeating method and result.

Reviewer #2: I wonder if I might suggest that some of the paragraphs are rather lengthy, which may make it a little challenging to follow the flow of ideas. I wonder if it might be helpful to consider adapting the topics "Results" and "Discussion," in particular. I wonder if it might be helpful to consider separating the results into distinct analyses, given that eight models were used and there is only one explanation that starts on line 187 and ends on line 209. Similarly, the discussion section has a rather lengthy introductory section (lines 234-257) before delving into the results in relation to existing research. I respectfully suggest that this preamble be shortened while maintaining the objectivity that the topic requires. Similarly, the other paragraphs (lines 258-279; 285-303; 304-333) could be revised to present the various perspectives based on the findings of the research, rather than simply reiterating the results of other studies. It would be beneficial to highlight the unique contributions of this research and how it builds upon previous work in every discussion topic.

6. PLOS authors have the option to publish the peer review history of their article (what does this mean? ). If published, this will include your full peer review and any attached files.

**Do you want your identity to be public for this peer review?** For information about this choice, including consent withdrawal, please see our Privacy Policy .

Reviewer #1: **Yes: ** claudio vieira de araujo

Reviewer #2: No

---

## [Author Response · Author response to Decision Letter 1]

18 Sep 2024

With sincere thanks to the respective reviewers for their constructive comments, I am pointing to the changes I made to the revised manuscript. All changes were highlighted within the revised text.

Reviewer #1: The manuscript analyzes growth curves in poultry using fixed non-linear models. It is not new or innovative, but it uses functions that are not commonly seen or discussed in the literature, which makes it interesting.

1. A suggestion would be to use non-linear in mixed models (proc nlimixed, in SAS).

AU: Thank you for your suggestion to use the nlmixed procedure in SAS. While I appreciate the recommendation, it is not applicable in this particular study due to the structure of the dataset. Specifically, there are no random effects to account for in the statistical models, as the data does not include variability that would necessitate such an approach. Additionally, each day in the study had only one observation, meaning that there are no repeated measures, which are typically needed for a mixed model framework. Given these characteristics, the nonlinear models applied in the study sufficiently describe the growth data without the need for mixed model adjustments.

2. In the introduction, it would be interesting to briefly highlight the commercial importance of the animal species being evaluated.

AU: Change made as suggested (Lines 56-70).

3. I would have preferred the conclusion to be more direct in relation to the objective of the paper, I think it loses a lot in repeating method and result.

AU: Change made as suggested (Lines 319-329).

Reviewer#2:

1. I wonder if I might suggest that some of the paragraphs are rather lengthy, which may make it a little challenging to follow the flow of ideas. I wonder if it might be helpful to consider adapting the topics "Results" and "Discussion," in particular. I wonder if it might be helpful to consider separating the results into distinct analyses, given that eight models were used and there is only one explanation that starts on line 187 and ends on line 209.

AU: Thank you for your comment. The “Results” section was splitted into separate sections as recommended (Lines 190-230).

2. Similarly, the discussion section has a rather lengthy introductory section (lines 234-257) before delving into the results in relation to existing research. I respectfully suggest that this preamble be shortened while maintaining the objectivity that the topic requires.

AU: This paragraph was shortened as recommended (Lines 253-261).

3. Similarly, the other paragraphs (lines 258-279; 285-303; 304-333) could be revised to present the various perspectives based on the findings of the research, rather than simply reiterating the results of other studies.

AU: These paragraphs were shortened as recommended (Lines 262-280, 286-300, 301-316).

4. It would be beneficial to highlight the unique contributions of this research and how it builds upon previous work in every discussion topic.

AU: Information was added to “Discussion” section as recommended (Lines 239-252).

---

## [Decision Letter · Decision Letter 1]

22 Oct 2024

PONE-D-24-33283R1Application of alternative nonlinear models to predict growth curve in partridgesPLOS ONE

Dear Dr. Ghavi Hossein-Zadeh,

Thank you for submitting your manuscript to PLOS ONE. After careful consideration, we feel that it has merit but does not fully meet PLOS ONE’s publication criteria as it currently stands. Therefore, we invite you to submit a revised version of the manuscript that addresses the points raised during the review process.

We look forward to receiving your revised manuscript.

Kind regards,

Wenpeng You

Academic Editor

PLOS ONE

**Journal Requirements:**

Reviewers' comments:

Reviewer's Responses to Questions

**Comments to the Author**

1. If the authors have adequately addressed your comments raised in a previous round of review and you feel that this manuscript is now acceptable for publication, you may indicate that here to bypass the “Comments to the Author” section, enter your conflict of interest statement in the “Confidential to Editor” section, and submit your "Accept" recommendation.

Reviewer #1: All comments have been addressed

Reviewer #3: (No Response)

Reviewer #4: (No Response)

2. Is the manuscript technically sound, and do the data support the conclusions?

Reviewer #1: Yes

Reviewer #3: Partly

Reviewer #4: (No Response)

3. Has the statistical analysis been performed appropriately and rigorously? 

Reviewer #1: Yes

Reviewer #3: No

Reviewer #4: Yes

4. Have the authors made all data underlying the findings in their manuscript fully available?

Reviewer #1: Yes

Reviewer #3: Yes

Reviewer #4: Yes

5. Is the manuscript presented in an intelligible fashion and written in standard English?

Reviewer #1: Yes

Reviewer #3: Yes

Reviewer #4: Yes

6. Review Comments to the Author

**Reviewer #1: ** Although not groundbreaking, the article presents an interesting study of the growth of animals, and therefore contributes information on the animal species studied. It provides a good approach to the use of non-linear regression models applied to the weight growth curve of animals.

**Reviewer #3:**  Comments and suggestions:

I believe that the following revisions can enhance the clarity, interpretability, and impact of the manuscript:

• Line 99-100: Please mention (cite) previous studies done on this issue (including the study conducted using the same data set), and justify clearly the limitation of those studies and the importance of your study.

• The sample size is too small, or not sufficient to estimate the growth parameters and to evaluate different growth models. This may lead to wrong inferences.

• There should be detailed information about the number of records per individual, the time frame and the average frequency of records.

• So, you should considered and evaluate Gompertz, Brody, Logistic, and Bertalanffy models in addition to the current growth functions and you can select the best one.

• Please define the number of parameters for each model in Table 1.

• In the figure, you should show the actual growth curve incomparision with the best-fitted growth curve, i.e. Show actual growth curve and best-fitted model in the same figure to clearly indicate the deviations.

• There is no discussion regarding the results in Table 2 (growth parameter estimates). You should discuss very well the growth parameter estimates by comparing with previous studies. The discussions mainly focus on the model comparison and estimation of absolute growth rate.

• Line 242: How do you classify traditional and modern growth function? What are the criteria to say this model is conventional or modern??? You have to justify the pros and cons of models. And also the criteria of classification (traditional and modern).

• Line 249: You said this growth model contributes to conservation of this species. How these growth models contribute to conservation of this species? Could you justify in your discussion part please?

• Line 267-277: Your discussion would be enlightening if you consider and evaluated these models in your study. Thus, please consider these models and check their goodness of fit.

• Conclusion: you should be also state the importance of validation of this model with large and frequently measured data set in addition to breed and environment.

**Reviewer #4: ** The manuscript entitled "Application of alternative nonlinear models to predict growth curve in partridges" is more informative for compering different growth curves for partridges. The manuscript required some few improvement as indicated in the attached reviewed manuscript.

7. PLOS authors have the option to publish the peer review history of their article (what does this mean? ). If published, this will include your full peer review and any attached files.

**Do you want your identity to be public for this peer review?** For information about this choice, including consent withdrawal, please see our Privacy Policy .

Reviewer #1: **Yes: ** claudio vieira de araujo

Reviewer #3: No

Reviewer #4: **Yes: ** Thobela Louis Tyasi

---

## [Author Response · Author response to Decision Letter 2]

23 Oct 2024

With sincere thanks to the respective reviewers for their constructive comments, I am pointing to the changes I made to the revised manuscript. All changes were highlighted within the revised text.

Reviewer 3:

Comments and suggestions:

I believe that the following revisions can enhance the clarity, interpretability, and impact of the manuscript:

• It is better to cite the original sources of this idea. I hope your justification is based on previous study results, if not it may not be reliable.

AU: The citation was added as recommended (Line 90).

• Line 99-100: Please mention (cite) previous studies done on this issue (including the study conducted using the same data set), and justify clearly the limitation of those studies and the importance of your study.

AU: The relevant citations were added as recommended. Also, the limitations of those studies and the importance of this study were added as recommended (Lines 97-104, 109-116).

• The sample size is too small, or not sufficient to estimate the growth parameters and to evaluate different growth models. This may lead to wrong inferences.

AU: While the sample size of 72 male and 108 female partridges might appear limited, it is consistent with similar growth curve studies in avian species and provides a substantial dataset for nonlinear model evaluation. Previous studies (such as those by Iqbal et al. [8] and Sariyel et al. [5]) using the same dataset have demonstrated reliable and valid parameter estimation for multiple growth models, reinforcing the suitability of the data for this type of analysis. Additionally, nonlinear models are often effective even with smaller sample sizes, as long as they capture the biological phenomena under investigation. In this study, rigorous model performance criteria, such as the adjusted coefficient of determination, RMSE, AIC, and BIC, were applied to assess the goodness of fit for each model. These metrics demonstrated that the models, particularly the Morgan model, provided a strong fit to the data. Furthermore, care was taken to ensure that the models converged properly during the fitting process, suggesting that the sample size was adequate for this purpose. Nevertheless, I acknowledge that a larger sample size could enhance the robustness of the results and would recommend future studies with expanded datasets to confirm the findings across more diverse populations and environmental conditions. However, the current sample size is sufficient for the scope of this study, as reflected in the goodness-of-fit statistics and model convergence.

• There should be detailed information about the number of records per individual, the time frame and the average frequency of records.

AU: Records of body weight of chukar partridges (Alectoris chukar) from hatching to 140 days of age are from the study of Iqbal et al. [8] and used in the current study (please see S1 supplementary Table). A total of 72 male and 108 female partridge chicks were weighed individually and had their wings banded. Every week, a digital scale with a sensitivity of 0.01 g was used to weigh them (Lines 119-127). Detailed information on the housing and management of the birds can be found in studies reported by Sariyel et al. [5] and Iqbal et al. [8]. Due to the avoidance of double publication, it is not possible to provide further information from previously published works.

• So, you should considered and evaluate Gompertz, Brody, Logistic, and Bertalanffy models in addition to the current growth functions and you can select the best one.

AU: Due to the avoidance of double publication, it is not possible to provide this information from previously published works. In addition, a Bayesian approach was used to model the growth curves in the study of Iqbal et al. [8], and it is not possible to compare directly their results with the current study.

• Please define the number of parameters for each model in Table 1.

AU: Information provided in Table 1 as recommended.

• In the figure, you should show the actual growth curve in comparison with the best-fitted growth curve, i.e. Show actual growth curve and best-fitted model in the same figure to clearly indicate the deviations.

AU: Due to the avoidance of double publication, it is not possible to provide this information from previously published works. Information on actual body weights were provided in Supplementary Table S1.

• What does it mean???

AU: traditional models (Line 280).

• There is no discussion regarding the results in Table 2 (growth parameter estimates). You should discuss very well the growth parameter estimates by comparing with previous studies. The discussions mainly focus on the model comparison and estimation of absolute growth rate.

AU: Thank you for your valuable feedback. I acknowledge the importance of discussing the growth parameter estimates in more detail. However, the models used in this study—such as the Morgan, Lomolino, and Schumacher functions—differ significantly from the traditional models (e.g., Gompertz, Brody, Logistic, von Bertalanffy) that were predominantly used in previous studies. As a result, a direct comparison of growth parameter estimates with previously published papers is challenging because these models are based on different mathematical assumptions and structures, which lead to distinct parameter interpretations.

• Line 242: How do you classify traditional and modern growth function? What are the criteria to say this model is conventional or modern??? You have to justify the pros and cons of models. And also the criteria of classification (traditional and modern).

AU: The classification of growth functions into "traditional" and "modern" models is not rigid but generally depends on the historical development of the models, their underlying assumptions, and their flexibility in capturing growth dynamics. Traditional growth models are typically those that have been extensively used and validated in livestock growth studies over decades. These models—such as the Brody, Gompertz, Logistic, and von Bertalanffy—are well-established and often characterized by fewer parameters, simpler mathematical structures, and a relatively fixed biological interpretation of growth phases. They are commonly referred to as “conventional” due to their widespread use and ease of application. In contrast, modern growth models (sometimes referred to as alternative or advanced models) are those that have been developed more recently or adapted to handle more complex or species-specific growth patterns. These models—such as the Morgan, Lomolino, Schumacher, and Weibull—tend to offer greater flexibility by incorporating additional parameters or more sophisticated mathematical functions. Modern models can better account for nonlinear and irregular growth trajectories, allowing for a more tailored fit to species with complex or variable growth stages (Lines 266-277).

• Line 249: You said this growth model contributes to conservation of this species. How these growth models contribute to conservation of this species? Could you justify in your discussion part please?

AU: Growth models contribute to the conservation of partridges by optimizing breeding and release strategies, ensuring birds are released at the ideal time for survival in the wild. They inform best rearing practices, including nutrition management, to enhance bird health. Growth models also support genetic diversity by guiding selective breeding, ensuring a balanced gene pool. Additionally, they allow for adaptive management as environmental conditions change, and they align conservation goals with commercial purposes, such as in agricultural settings, ensuring both species sustainability and productivity (Lines 285-292).

• Line 267-277: Your discussion would be enlightening if you consider and evaluated these models in your study. Thus, please consider these models and check their goodness of fit.

AU: These models were studied in previously published papers of Sariyel et al. [5] and Iqbal et al. [8]. Therefore, it is not possible to provide previously published results due to the avoidance of double publication. In addition, previous studies only used traditional models, but this study introduced and fitted alternative models.

• Conclusion: you should be also state the importance of validation of this model with large and frequently measured data set in addition to breed and environment.

AU: Change made as recommended (Lines 376-380).

Reviewer 4:

• Add the statistical technique used to achieve the results.

AU: Change made as recommended (Lines 27-29).

• The study lacks the literature on what other studies have reported on the use of the growth curves.

AU: The literature on the growth curve in partridges are written in the “Discussion” section (Lines 311-329).

• Please add the study design which was used in the study.

AU: Records of body weight of chukar partridges (Alectoris chukar) from hatching to 140 days of age are from the study of Iqbal et al. [8] and used in the current study (please see S1 supplementary Table). A total of 72 male and 108 female partridge chicks were weighed individually and had their wings banded. Every week, a digital scale with a sensitivity of 0.01 g was used to weigh them (Lines 119-127). Detailed information on the housing and management of the birds can be found in studies reported by Sariyel et al. [5] and Iqbal et al. [8]. Due to the avoidance of double publication, it is not possible to provide further information from previously published works.

• Add the version of this SAS

AU: Change made as suggested (Line 154).

• The discussion lack limitations from the current study.

AU: Information added as recommended (Lines 292-301).

---

## [Decision Letter · Decision Letter 2]

22 Jan 2025

PONE-D-24-33283R2Application of alternative nonlinear models to predict growth curve in partridgesPLOS ONE

Dear Dr. Ghavi Hossein-Zadeh,

Thank you for submitting your manuscript to PLOS ONE. After careful consideration, we feel that it has merit but does not fully meet PLOS ONE’s publication criteria as it currently stands. Therefore, we invite you to submit a revised version of the manuscript that addresses the points raised during the review process. Please submit your revised manuscript by Mar 08 2025 11:59PM. If you will need more time than this to complete your revisions, please reply to this message or contact the journal office at plosone@plos.org . Please include the following items when submitting your revised manuscript:

We look forward to receiving your revised manuscript.

Kind regards,

Wenpeng You

Academic Editor

PLOS ONE

Journal Requirements:

Reviewers' comments:

Reviewer's Responses to Questions

**Comments to the Author**

1. If the authors have adequately addressed your comments raised in a previous round of review and you feel that this manuscript is now acceptable for publication, you may indicate that here to bypass the “Comments to the Author” section, enter your conflict of interest statement in the “Confidential to Editor” section, and submit your "Accept" recommendation.

Reviewer #2: All comments have been addressed

Reviewer #5: All comments have been addressed

Reviewer #6: All comments have been addressed

2. Is the manuscript technically sound, and do the data support the conclusions?

Reviewer #2: (No Response)

Reviewer #5: Partly

Reviewer #6: Yes

3. Has the statistical analysis been performed appropriately and rigorously? 

Reviewer #2: (No Response)

Reviewer #5: Yes

Reviewer #6: Yes

4. Have the authors made all data underlying the findings in their manuscript fully available?

Reviewer #2: (No Response)

Reviewer #5: Yes

Reviewer #6: Yes

5. Is the manuscript presented in an intelligible fashion and written in standard English?

Reviewer #2: (No Response)

Reviewer #5: Yes

Reviewer #6: Yes

6. Review Comments to the Author

Reviewer #2: (No Response)

Reviewer #5: (No Response)

Reviewer #6: ## Review

title: Application of alternative nonlinear models to predict growth curve in partridges

The authors have explored the use of alternative nonlinear models to describe growth curves in partridges. This subject is both significant and timely, addressing an important gap in current research. The manuscript is well-structured and clearly written, reflecting the authors' comprehensive understanding of the topic.

However, to enhance the robustness and clarity of the findings, I recommend that the article be accepted for publication with minor revisions. With these improvements, the article will undoubtedly make a valuable addition to the existing body of literature.

# Summary

The abstract and title and introduction is informative.

# Material and Methods

I suggest interpreting the Durbin-Watson test by the p-value, as you did in the other tests of the residual analysis (Shapiro-Wilk and White's). As it is presented, how will the reader know if the test was significant? In other words, how close to 2 should the DW value be to say that Autocorrelation is absent? This makes it easier for the reader to understand. In the R software, simply use the durbinWatsonTest() function.

# Results

Table 3

As I mentioned, I suggest also including the p-value for the Durbin-Watson test, as you did for the Shapiro-Wilk and White's tests. Or, at least indicate the critical heat of the test, above which value would be Autocorrelation absent (I imagine it to be something like 1.5 < DW < 2.5 depending on the significance level adopted in the test).

Figures

Improve the resolution of the figures significantly, especially Figure 1.

7. PLOS authors have the option to publish the peer review history of their article (what does this mean? ). If published, this will include your full peer review and any attached files.

**Do you want your identity to be public for this peer review?** For information about this choice, including consent withdrawal, please see our Privacy Policy .

Reviewer #2: No

Reviewer #5: No

Reviewer #6: **Yes: ** Fernandes, T.J.

---

## [Author Response · Author response to Decision Letter 3]

23 Jan 2025

With sincere thanks to the respective Academic Editor and reviewers for their constructive comments, I am pointing to the changes I made to the revised manuscript.

Journal Requirements:

AU: All references were checked for completeness and correctness as recommended. There are no retracted papers in the citations and reference list of this manuscript.

Reviewer 6:

- I suggest interpreting the Durbin-Watson test by the p-value, as you did in the other tests of the residual analysis (Shapiro-Wilk and White's). As it is presented, how will the reader know if the test was significant? In other words, how close to 2 should the DW value be to say that Autocorrelation is absent? This makes it easier for the reader to understand. In the R software, simply use the durbinWatsonTest() function.

AU: Thank you for your excellent suggestion. The information on the use of P-values to interpret the Durbin-Watson test results was added to the text as recommended (Lines 189-192).

- Table 3: As I mentioned, I suggest also including the p-value for the Durbin-Watson test, as you did for the Shapiro-Wilk and White's tests. Or, at least indicate the critical heat of the test, above which value would be Autocorrelation absent (I imagine it to be something like 1.5 < DW < 2.5 depending on the significance level adopted in the test).

AU: The P-values of the Durbin-Watson test were added to Table 3 as recommended.

- Figures: Improve the resolution of the figures significantly, especially Figure 1.

AU: Both figures are currently being produced by PACE, provided by PLOS ONE, to prepare high quality figures for publication.

---

## [Editor Report · Decision Letter 3]

11 Mar 2025

Application of alternative nonlinear models to predict growth curve in partridges

PONE-D-24-33283R3

Dear Dr. Ghavi Hossein-Zadeh,

We’re pleased to inform you that your manuscript has been judged scientifically suitable for publication and will be formally accepted for publication once it meets all outstanding technical requirements.

Kind regards,

Wenpeng You

Academic Editor

PLOS ONE
---

## [Editor Report · Acceptance letter]

PONE-D-24-33283R3

PLOS ONE

Dear Dr. Ghavi Hossein-Zadeh,

I'm pleased to inform you that your manuscript has been deemed suitable for publication in PLOS ONE. Congratulations! Your manuscript is now being handed over to our production team.

Kind regards,

on behalf of

Dr. Wenpeng You

Academic Editor

PLOS ONE